# Cell Therapy of Stroke: Do the Intra-Arterially Transplanted Mesenchymal Stem Cells Cross the Blood–Brain Barrier?

**DOI:** 10.3390/cells10112997

**Published:** 2021-11-03

**Authors:** Konstantin N. Yarygin, Daria D. Namestnikova, Kirill K. Sukhinich, Ilya L. Gubskiy, Alexander G. Majouga, Irina V. Kholodenko

**Affiliations:** 1Laboratory of Cell Biology, Orekhovich Institute of Biomedical Chemistry, 119435 Moscow, Russia; irkhol@yandex.ru; 2Pirogov Russian National Research Medical University of the Ministry of Healthcare of the Russian Federation, 117997 Moscow, Russia; dadnam89@gmail.com (D.D.N.); gubskiy.ilya@gmail.com (I.L.G.); 3Radiology and Clinical Physiology Scientific Research Center, Federal Center of Brain Research and Neurotechnologies of the Federal Medical Biological Agency, 117513 Moscow, Russia; 4Laboratory of Problems of Regeneration, Koltzov Institute of Developmental Biology of the Russian Academy of Sciences, 119334 Moscow, Russia; sukhinichkirill@gmail.com; 5D. Mendeleev University of Chemical Technology of Russia, 125047 Moscow, Russia; alexander.majouga@gmail.com

**Keywords:** mesenchymal stem cells, cell therapy, stroke, blood–brain barrier, neurovascular unit, stroke models

## Abstract

Animal model studies and first clinical trials have demonstrated the safety and efficacy of the mesenchymal stem cells’ (MSCs) transplantation in stroke. Intra-arterial (IA) administration looks especially promising, since it provides targeted cell delivery to the ischemic brain, is highly effective, and can be safe as long as the infusion is conducted appropriately. However, wider clinical application of the IA MSCs transplantation will only be possible after a better understanding of the mechanism of their therapeutic action is achieved. On the way to achieve this goal, the study of transplanted cells’ fate and their interactions with the blood–brain barrier (BBB) structures could be one of the key factors. In this review, we analyze the available data concerning one of the most important aspects of the transplanted MSCs’ action—the ability of cells to cross the blood–brain barrier (BBB) in vitro and in vivo after IA administration into animals with experimental stroke. The collected data show that some of the transplanted MSCs temporarily attach to the walls of the cerebral vessels and then return to the bloodstream or penetrate the BBB and either undergo homing in the perivascular space or penetrate deeper into the parenchyma. Transmigration across the BBB is not necessary for the induction of therapeutic effects, which can be incited through a paracrine mechanism even by cells located inside the blood vessels.

## 1. Introduction

Ischemic stroke incidence and mortality rate, as well as the proportion of post-stroke patients remaining disabled stay at an unacceptably high level worldwide [1]. Currently, there are just two effective and internationally recommended treatment methods for acute ischemic stroke: intravenous thrombolysis with tissue-type plasminogen activator (t-PA), and endovascular mechanical thrombectomy [2]. These methods have contraindications and are limited by a narrow “therapeutic window”, which according to the latest data does not exceed 9 h for systemic thrombolysis [3] and 24 h for endovascular thrombectomy [4]. The efficacy of the existing approaches to the post-stroke rehabilitation is mostly insufficient as well. As a result, the majority of stroke survivors suffer from life-long neurological deficits due to the formation of cerebral infarction and unsatisfactory structural and functional regeneration of the not fatally damaged brain tissue [5]. Finding new approaches to the development of an effective stroke therapy at both acute and chronic stages of the disease is a task of utmost urgency and importance. Transplantation of different types of stem cells has the potential to develop into an effective method of stroke therapy [6,7]. Specifically, over the past two decades, multiple basic studies and clinical trials have demonstrated that transplantation of the mesenchymal stem cells (mesenchymal stromal cells, MSCs) is safe [8] and can promote recovery after stroke (reviewed in [9,10]). According to the International Society of Cell Therapy criteria, MSCs are defined as multipotent non-hematopoietic stem cells that are adherent to plastic and can differentiate into adipocytes, chondroblasts, osteoblasts, and myocytes in vitro. MSCs are also required to have proper CD phenotype: manifest expression of CD105, CD73, and CD90 and lack or low expression of CD45, CD34, CD14 or CD11b, CD79α or CD19, and HLA-DR [11]. Low or nonexistent expression of HLA-DR is considered one of the therapeutic advantages of MSCs, since, presumably, it facilitates transplantation of allogenic and even xenogenic cells without immunosuppression [12]. MSCs are thought to have other advantages, such as availability and low tumorigenicity [13]. MSCs can be obtained from the majority of tissues, including bone marrow, adipose tissue, skin, internal organs, placenta, cord blood, amniotic fluid, and many others [14,15,16,17]. Notably, MSCs obtained from different sources or from several individual donors can have certain differences in genome expression and, as a result, varying properties [18,19,20].

To introduce transplantation of MSCs into clinical practice, it is necessary to understand the mechanisms of their therapeutic action better. Several mechanisms underlying the therapeutic effects of MSCs have been proposed, among them paracrine action, direct interaction with other cells, and substitution of dead/damaged parenchymal cells after homing inside the tissue and differentiation [9,16]. Paracrine action means secretion of factors inducing activation or modulation of activity of tissue stem/progenitor cells, stem cell niche cells, immune cells, epithelial cells, and perivascular cells. Direct interaction of transplanted MSCs with host cells is a rather recently discovered alternative mechanism [13]. The ability of MSCs to transdifferentiate into mature neurons after transplantation and substitute damaged brain cells is debatable, to say the least. However, some authors have reported that, after intracerebral administration, some of the injected MSCs lost the expression of MSC markers and acquired neuronal phenotype [21]. In vitro, it has been shown that MSCs can transdifferentiate into cells of endodermal origin—such as hepatocytes [22,23] and β-cells of the pancreatic islets [24]. During the last two decades, many studies attempting to achieve neuronal transdifferentiation of MSCs have been conducted. Several papers reported successful transdifferentiation into neuron-like cells, but in fact the obtained cells were not fully functional neurons (discussed in detail by Choudhary et al. [25]).

While the direct replacement of damaged neural cells by transplanted MSCs after stroke is debatable, their paracrine effects in the same pathology are well established. MSCs secrete a wide range of different factors and extracellular vesicles that may have an important role in the regeneration of brain tissue and restoration of brain functioning [8]. Anti-inflammatory and immunomodulatory paracrine activities of MSCs are very well documented [9,26]. In addition, recent studies have demonstrated direct cell-to-cell interactions and the possibility of transfer of mitochondria and cytoplasm components from transplanted MSCs to host neural cells [27,28]. Recently, a new mechanism by which MSCs could mediate their positive effects was reported (discussed by Cheung et al. [29]). In a graft-versus-host disease murine model it was demonstrated that transplanted MSCs’ contact with the recipient’s cytotoxic cells has dramatic consequences for both. MSCs undergo in vivo apoptosis and are phagocytosed by host immune cells, while the latter go through complete transformation of their immune properties, leading to the attenuation of the disease symptoms [30]. Whether the described scenario may be in place in the case of MSC-based cell therapy of stroke needs further investigation. Despite extensive investigations, the exact cellular and molecular mechanisms underlying the therapeutic properties of MSCs are yet to be fully disclosed.

Several routes of MSC transplantation in stroke, including the intra-arterial (IA), intravenous (IV), intraperitoneal, intracerebral, intraventricular, intrathecal, and intranasal delivery were tried and showed therapeutic efficacy [10,31]. However, the best delivery route, as well as the ‘therapeutic window’ duration have not been so far agreed upon. IA administration seems to be one of the most promising, since it provides targeted cell delivery to the ischemic brain, bypassing blood filtering organs (lungs, liver, spleen, etc.), is highly effective, and can be safe as long as the infusion parameters (cell dose, infusion velocity, and others) are maintained within appropriate limits [32,33,34]. Actually, due to rapid development of the endovascular treatment methods, IA delivery has become minimally invasive and can be easily adapted for clinical practice [35]. Immediately after IA transplantation, MSCs get into the cerebral arterial system and come into contact with the inner walls of blood vessels. Tracking their fate, including interactions with the components of the blood–brain barrier (BBB) and translocations, is crucially important and could help to establish the mechanisms of the positive therapeutic effects of MSCs [36].

After IA transplantation, the substitution of the damaged cells by their counterparts differentiating from transplanted MSCs is only possible if the latter penetrate from blood into the brain tissue and undergo homing and differentiation. Direct cell–cell interaction of transplanted MSCs with neurons and glia also requires passing the BBB. On the other hand, paracrine effects can be induced even by cells remaining inside the circulation.

In this review, we analyze the available data concerning interactions of MSCs with the BBB components and their ability to cross BBB in vitro and in the experimental stroke models in vivo after IA administration. Analyzing publications describing the in vivo studies, we concentrated on the search of the publications presenting data clearly showing the location of specifically labeled and positively traceable transplanted MSCs within blood vessels or in brain parenchyma. These papers were regarded as the most reliable source of information about the interactions of MSCs with the BBB, the neurovascular unit, and their components, and about the ability of MSCs to cross the BBB. Other works were also considered, but with some reservation.

## 2. Blood–Brain Barrier in Stroke and Its Response to MSC Transplantation

The BBB separates brain parenchyma from the bloodstream and at the same time supports two-way communication between them. It is a unique border consisting of endothelial cells, basement membrane, pericytes, and astrocytes [37]. The blood–tissue barrier in the central nervous system is quite different from the other blood–tissue barriers. About 98% of prospective medications for brain disorders fail to penetrate BBB, suggesting its unusually high selectivity [38]. Recently a new, broader concept of the neurovascular unit (NVU) was introduced in neuroscience [39]. The NVU includes all BBB components and also neurons, microglia and other glial cells, smooth muscle cells and extracellular matrix. All NVU components are functionally combined to form a single system that provides maintenance of the BBB and the exchange of information and signals between neurons, cerebral vasculature, and blood, thus ensuring cerebral homeostasis and control of the cerebral blood flow (reviewed in [40]).

Damage occurring to the BBB and the NVU plays an important role in the pathogenesis of stroke (comprehensively reviewed in [41]). The permeability of the BBB depends on the stage of the disease (reviewed in detail by [42]), as well as of stroke type and severity and individual characteristics of the patient or experimental animal [43,44,45,46]. In general, the early opening of the BBB occurs within the first 6 h after the stroke onset, in the hyperacute stage, when ischemia leads to an avalanche of pathological reactions [47], including cytotoxic edema, disruption of tight junctions mostly due to the effect of matrix metalloproteinase-2 [48], and damage of varying severity to brain cells, including the elements of NVU. This first stage of the BBB opening can be reversible in case of restoration of the cerebral blood flow [46]. Subsequent cell death and the NVU dysfunction leads to the development of vasogenic edema, infiltration of different subpopulations of immune cells (predominantly macrophages and neutrophils) into the brain parenchyma [41], and development of neuroinflammation [49,50]. Invading neutrophils secrete matrix metalloproteinases (MMPs), especially MMP-9, that contribute to further damage of tight junctions and increased vascular permeability [51,52]. The second peak of the BBB permeability is observed in the acute stage of stroke with a maximum between 48–96 h after the onset of ischemia, is irreversible and leads to increased intracranial pressure [51,53,54,55]. The disruption of the BBB and severe endothelial and basal membrane dysfunction in the ischemia core lead to the extravasation of blood, impregnation of the brain parenchyma with its components, and increased risk of the hemorrhagic transformation of the ischemic infarction zone [56]. This risk may be amplified by the reperfusion therapy [42,57]. The elevated BBB permeability persists in the subacute stage for 1–3 weeks and, to a lesser extent up to 6 weeks after stroke, spreading to its chronic period [42]. Prolonged BBB disfunction is believed to be associated with incomplete BBB recovery [58] due to activation of neuro- and angiogenesis [59,60]. Indeed, though the incidence of intracranial hemorrhage in stroke patients is highest during the first 30 days after stroke, it still remains higher than in the general population for at least one year thereafter [61]. Along with the negative aspects, the enhanced BBB permeability has positive consequences facilitating the delivery of therapeutic agents to the brain [42].

Though the NVU itself exerts protective effects through secretion of a variety of neurotrophic factors and modulation of the inflammatory immune response [62,63], its own regeneration capacity for recovery after ischemic stroke is limited. There is growing evidence that MSC-based cell therapy can decrease BBB disruption and promote repair of the NVU (comprehensively reviewed in [64] and the molecular mechanism thoroughly discussed in [51]). It was shown that in rats the systemic administration of MSCs promoted BBB stabilization due to the reduction of BBB leakage and enhancement of microvascular repair [65]. MSCs transplantation can decrease the expression of IL-1b, IL-6, and TNF-a, and upregulate the tight junction proteins ZO-1 and Claudin-5, thus contributing to the reduction of the BBB permeability in rat stroke model [66]. Huang et al. demonstrated that MSCs induced decline of the tight junction proteins degradation, as well as increased expression of CD31 and pericyte density in the NVU after intravenous administration to rats [67]. Chung et al. [68] has also shown that systemic administration of MSCs into rats provides a protective effect against hippocampal neuronal death due to reduction of endothelial damage and neutrophil infiltration. Moreover, after IV infusion of MSCs in the model of small vessel disease in stroke-prone spontaneously hypertensive rats, Nakazaki et al. [69] observed restoration of BBB function via activation of both transforming growth factor-β and angiopoietin 1 signaling pathways, leading to remodeling of the microvasculature. Notably, in all described studies the structural restoration of the BBB came along with its functional recovery. There are studies reporting that transplantation not only of MSCs, but also of MSC-conditioned culture medium in rat stroke model [70] or MSC-derived extracellular vesicles (reviewed by [71]) reduced the destruction of BBB and provided the NVU repair.

Recently, it was shown that MSCs can promote their beneficial effects via restoration of the ‘glymphatic’ system. Glial components (end-feet of astrocytes), which are functionally active components of the NVU and provide the outer border of the perivascular space from the side of the brain parenchyma, perform the functions of the central nervous system cleansing and waste exchange, forming the so-called “glymphatic” system [72]. Aquaporin-4 (AQP4) plays a major role in the functioning of the “glymphatic” system [73]. Upregulation of AQP4 occurs in a variety of brain injuries, including stroke [74]. Suppression of AQP4 upregulation after brain injury has a beneficial therapeutic effect [75]. In a model of cerebral ischemia in mice, it was shown that intracranial injection of rat BM-MSCs led to a decrease in astrocyte apoptosis and inhibited AQP4 upregulation [76], probably reflecting another protective mechanism underlying the maintenance/restoration of the BBB by MSCs. In their recent immunohistochemical study of postmortem human brain samples, Mezey et al. [77] showed that the perivascular space contains endothelial cells expressing the same markers as peripheral lymphatic endothelial cells such as LYVE1, PDPN, VEGF3, and Prox1. They also found that CD3-positive T cells are located in close proximity to these endothelial cells. Finding the expression of ICAM1 on endothelial cells expressing lymphatic markers, and the expression of the LFA1 integrin on T cells in brain tissues, the authors suggested that T cells can move in these spaces in the same way as they ‘roll’ along the endothelial cells of peripheral vessels. This can lead to a slow ‘flow’ of immune cells along the nerves into the perivascular space. Thus, the perivascular network probably allows not only solutes and metabolites, but also cells to move extracranially from brain parenchyma, ultimately to the venous sinuses and the peripheral lymphatic system. One can speculate about the existence of a similar mechanism of migration of transplanted MSCs in the perivascular space. Potentially, this mechanism could provide one of the ways to remove transplanted cells from the brain. However, this issue requires further careful study.

Another possible mechanism underlying MSC-mediated BBB repair is their ability to regulate the activity of matrix metalloproteinases ([78,79] and reviewed by [51]). After ischemic brain damage, transmigrating leukocytes—in particular neutrophils—produce matrix metalloproteinases (MMP), for example, MMP-9, which provoke the development of neuroinflammation and aggravate the BBB integrity interruption. Despite the fact that MSCs themselves produce various MMPs, their activity is controlled by the production of the tissue inhibitor of metalloproteinase (TIMP) [80]. It has been shown that MSCs are capable of inhibiting exogenous MMPs, namely MMP-2 and MMP-9, through TIMP-2 and TIMP-1, respectively [81]. Thus, inhibition of MMP9 in ischemic brain injury can serve as an effective therapeutic target for transplanted MSCs, preventing neuroinflammation and BBB compromise. Nevertheless, the mechanisms mediating positive effects of MSCs on BBB restoration need further investigation.

## 3. In Vitro Studies of the MSC Transmigration through the BBB

To transmigrate from blood into the brain parenchyma, MSCs have to overcome four sequential roadblocks: endothelium, the basement membrane, the vascular smooth muscle cells and the pericyte layer. Despite ongoing efforts to create in vitro models of the BBB, including components other than endothelium [82,83,84,85], a simplified model in the form of a monolayer of brain microvascular endothelial cells (BMECs) is still commonly used to study the transmigration of MSCs across the BBB. The major events happening in the course of the transendothelial migration of MSCs are in some ways similar to those occurring to leukocytes [86,87]. However, the molecular mechanisms involved in the MSC transmigration across the BBB remain not fully understood.

The structure of the BBB, which separates the microenvironment of the brain from the circulating blood, differs significantly from that of the peripheral vessel walls [88]. Though BBB is integrated into the NVU, and endothelium is just a part of BBB, endothelial cells still are the main cellular component of the BBB regulating how permeable it is. Brain microvascular endothelial cells (BMECs) are significantly different from the vascular endothelium of other organs. They are characterized by the ability to form tight junctions, low number of pinocytic vesicles, and the presence of specialized transport systems responsible for maintaining the ionic and metabolic homeostasis of the brain parenchyma. In addition to endothelial cells, pericytes are the structural and functional components of the BBB, also determining its integrity and permeability [89]. While pericytes line the capillaries, larger vessels such as arteries, arterioles, venules, and veins are lined with the vascular smooth muscle cells. Both of these types of cells make up mural cells that support blood vessels [90]. Currently, no specific markers have been found to distinguish between pericytes and smooth muscle cells; nevertheless, new in vivo labeling technologies can solve this problem to some extent [91]. Vascular smooth muscle cells are an important component of the NVU controlling cerebral blood flow and maintaining the BBB integrity. It has been shown that in ischemic stroke cerebral vascular smooth muscle cells can switch between detrimental phenotype [92] and beneficial phenotype, participating in the processes of brain repair [93].

One of the main questions concerning the mechanisms of MSCs transmigration across the BBB is whether the route of their passage through the endothelial layer is paracellular or transcellular. The paracellular transmigration involves the transfer of cells through the intercellular space between endothelial cells due to the disruption of tight junctions and the temporary formation of ‘gaps’, while the transcellular pathway is the penetration of cells directly through individual endothelial cells [94]. Leukocytes use both ways of diapedesis across the BBB [95], while tumor cells that metastasize to the brain usually prefer only one of the two ways of transendothelial migration, depending on the type of tumor. For example, Herman et al. [96] showed that melanoma cells predominantly use the paracellular way of transendothelial migration, while breast cancer cells migrate across the brain endothelial layer via the transcellular pathway.

The interaction of MSCs with the endothelium can occur in three ways. In the in vitro system utilizing adult human lung (hLMVECs) and cardiac (hCMVECs) microvascular endothelial cells, 50% of human bone marrow MSCs integrated into the endothelial layer, inducing endothelial pocketing, rather than transmigrated across it [97]. In addition to integration, MSCs, similar to leukocytes, also used two pathways to cross the endothelial barrier. In most diapedesis events, MSCs migrated across the paracellular gaps formed in the sites of local destruction of adhesive and tight junctions. With a lower frequency (~20–30% of diapedesis events), MSCs migrated directly through individual endothelial cells due to the de novo formation of transcellular pores in endothelial cells [97].

In the BBB models employing brain endothelial cells, the integration of MSCs was observed very rarely, and transmigration almost always occurred along the paracellular pathway. Using rat bone marrow MSCs and human BMECs, Lin et al. [98] found that MSCs transmigrate across the endothelial monolayer and detected the transmigration route. In this study, incubation of human BMEC monolayer with rat MSCs led to an obvious disorder in the distribution of the ZO-1 tight junction protein and to changes in another tight junction protein, occluding, distribution with its transition from the detergent-insoluble to the detergent-soluble state. This obviously reflected the rupture of the integrity of tight junctions, and, as a consequence, an increase in the paracellular permeability of brain endothelium. Using a time-lapse imaging technique, Matsushita et al. [99] visualized the migration of rat MSCs across the monolayer of rat BMECs through the intercellular spaces between BMECs. They also measured transendothelial electrical resistance and found its decrease indicated decline in the barrier function and substantiated the paracellular mechanism of the transendothelial migration of MSCs.

Several possible mechanisms of the induction of transient gaps formation between BBB endothelial cells by MSCs in vitro have been described. Thus, it was shown that medium conditioned by MSCs cultured under hypoxic conditions and containing high levels of VEGF and matrix metalloprotease-9 (MMP-9), induced a sharp increase in the permeability of the BMEC monolayer [100]. The increase of the BMEC monolayer permeability was partially inhibited by the anti-VEGF antibodies and a MMP-9 inhibitor suggesting direct involvement of VEGF and MMP-9 in the disruption of tight junctions and the formation of gaps. This conclusion is supported by several findings.

It has been reliably proven that VEGF is one of the key factors enhancing BBB permeability under ischemia/hypoxia conditions [101,102]. VEGF-induced increase in vascular permeability begins with its interaction with the VEGF receptor and is further mediated by intracellular signal transduction cascades involving two non-receptor tyrosine kinases: the proto-oncogene tyrosine-protein kinase Src and the focal adhesion kinase Fak. Src is phosphorylated in response to VEGF. In turn, activated Src regulates the expression and/or stability of tight junction proteins. Src deficiency or blockade of Src phosphorylation results in abnormal vascular permeability in response to VEGF [103]. Claudin-5 is a tight junction protein and an important structural element of the BBB mediating the selective BBB permeability. In claudin-5 knockout mice, the BBB is opened [104]. In human brain endothelial cells, the expression of this protein is dynamically regulated by VEGF at the transcriptional level. Direct injection of VEGF into the cerebral cortex of mice downregulates claudin-5 and induces BBB damage [105]. It is noteworthy, that inhibition of Src phosphorylation by its specific inhibitor PP2 preserves the expression of claudin-5 and attenuates the destruction of the BBB in the ischemic mouse brain [106]. Fak, an integrin-activated kinase, also mediates VEGF-induced vascular permeability. Genetic or pharmacological inhibition of Fak activity in endothelial cells blocks VEGF-induced permeability mediated by VEGF receptors and Src activation in vivo [107].

MMP-9 plays an important role in disrupting the integrity of the BBB and, as a consequence, increasing its permeability. It has been shown that the level of MMP-9 increases significantly in ischemic, hemorrhagic, and traumatic brain injuries [108]. Inhibition of MMP-9, in contrast to inhibition of MMP-2, the level of which also increases after brain injury, leads to a rapid decrease in the BBB permeability [109] indicating that MMP-9 is the dominant protease that negatively affects the BBB under brain damage conditions [110]. As mentioned above, MSCs are able to regulate the activity of the MMPs that they produce, as well as exogenous MMPs. Thus, on one hand, MSCs, using their own secreted MMPs, facilitate their penetration through the BBB; however, on the other hand, they suppress the activity of exogenous MMPs, thereby contributing to the restoration of the BBB.

The so-called “zipper mechanism” may be another mechanism underlying the formation of temporary gaps between BBB endothelial cells in the process of their interaction with MSCs. It has been described for human embryonic stem cells-derived MSCs (hES-MSCs) and its essence is as follows. Since both BMECs and hES-MSCs express the same cell adhesion molecules CLN-5, ZO-1, and occluding, their co-cultivation may result in the formation of temporary homophilic and/or heterophilic interactions between the corresponding proteins and formation of tight/adhesive junctions between MSCs and endothelium [111]. This process may be competing with homophilic interactions between these proteins in tight junctions between endothelial cells leading to their destruction and, as a consequence, to temporary opening of the gaps and migration of MSCs across them. A similar mechanism could not have been shown for bone marrow MSCs, presumably since they do not express any of the above proteins [111].

It has been shown that the main signaling pathways involved in the processes of cytokine-directed migration of MSCs are PI3K/Akt [112,113], Rho/ROCK [114], and PKC [115,116]. Lin et al. [98], using inhibitory analysis, showed that PI3K and ROCK signaling pathways—but not the PKC signaling pathway—are involved in the process of transendothelial migration of MSCs. It is noteworthy that inhibition of PI3K in MSCs prevented the disruption of tight junctions between brain endothelial cells induced by MSCs and reduced the rate of transendothelial migration of MSCs, whereas inhibition of ROCK in MSCs enhanced disassembly of tight junctions and transendothelial migration through BMEC [98]. On the other hand, Feng et al. [117] showed an important role of the chemokine CXCL11 and its receptor CCR3 in increasing the BBB permeability and, consequently, enhancing the transmigration of MSCs and determined that ERK1/2 is the key signaling pathway in this process. The authors showed that ERK1/2 inhibitor PD98059 blocked CXCL11-induced increase in HRP flux, while ROCK (Y27632), PI3K (LY294002), and PKC (Gö6976) inhibitors had no effect on BMEC monolayer permeability [117].

Sadly, until now no published papers have reported any results of the in vitro experiments describing the transmigration of MSCs across complex models of the BBB comprising not just BMEC monolayer, but other BBB components. Accordingly, there is no complete picture reflecting the mechanisms of MSC transmigration, and the key regulators of this process have been only partly identified. There is urgent need for the application of more complete and relevant in vitro BBB models to MSC transmigration research.

## 4. Transmigration of the Intra-Arterially Transplanted MSCs cross the BBB in Experimental Stroke

Studying the extravasation of MSCs systemically transplanted into animals with experimental stroke is essential for understanding the mechanisms underlying the beneficial effects of cell therapy in this condition. Despite proven therapeutic efficacy of the IV MSC transplantation [118], the number of cells reaching the brain after IV injection is small [119,120,121] or even negligible [122,123]. This limits the possibility to assess the details of the process of extravasation of MSCs after the IV administration. Infusion of cells directly into the brain arterial system provides targeted cell delivery to the brain vasculature bypassing blood filtering organs [124] and ensures the first passage of transplanted cells through brain capillaries, not lung, liver, or other peripheral organ capillaries. Therefore, IA administration is more adapted to studying the MSC’s transmigration across the BBB.

The experimental assessment of this process is not an easy task. Part of the published papers present results based on insufficiently reliable methods of identification of transplanted cells. One more problem is that the diversity of the employed experimental approaches impeding the estimation and comparison of the results. Analyzing the literature, we aimed to select studies where the exact location and fate of MSCs after transplantation were clearly revealed and illustrated. The results of these relatively few studies are presented in Table 1. Besides the ability of cells to cross the BBB, the following indicators and parameters were accounted for and presented in Table 1: stroke model, animal species, type of MSCs, dose, delivery route and time window, use of immunosuppression, thromboembolism control, therapeutic effects of MSCs’ transplantation. In all studies, stroke modeling was performed in rats, and in most cases cerebral ischemia was induced by transient occlusion of the middle cerebral artery. Allogenic or xenogenic MSCs largely derived from bone marrow were transplanted in the hyperacute and acute periods of stroke (from 30 min to 7 days after the brain ischemia onset). In this time window, as mentioned above, the high permeability of the BBB may facilitate the entrance of transplanted MSCs from circulation into the brain [42]. In all studies, cells were transplanted through the internal carotid artery or the stump of the external carotid artery.

The papers listed in Table 1 also met a number of additional criteria, though not all additional criteria in each paper. IA administration of MSCs can cause a number of complications, which can be prevented. In vitro MSCs form highly adhesive cultures consisting of cells of relatively large sizes between 13 and 30 μm depending on MSC origin, stage of cell cycle, culture method and passage number [125,126]. In suspension MSCs become rounded, but tend to form aggregates. Consequently, IA administration of MSCs can potentially cause microembolism and formation of secondary thromboembolic strokes [122,127]. In addition, such parameters as cell dose, infusion speed, treatment window, and preservation of the blood circulation during the infusion of the cell suspension are very important factors affecting the safety of transplantation [32]. In the chosen studies, the cell dose varied between 1 × 10^4^ to 2 × 10^6^, and infusion parameters were slightly different from study to study. In Table 1, we highlighted the presence or absence of the thromboembolism control, because we consider this factor crucial for correct interpretation of research results. Without adequate control of thromboembolism MSC entrapment in cerebral vasculature causing vessel blockade and dilatation can be overlooked [122]. It should be also mentioned, that in different studies the researchers used varying anesthesia protocols and probably unspecified medications to keep rats in better condition during and after surgery, and this might potentially affect the permeability of the BBB [128,129,130,131,132]. Moreover, a recent study of Schaffenrath et al. [133] has demonstrated the presence of some differences in gene expression in brain endothelial cells between different strains of laboratory rodents that potentially may have an impact on the interactions of MSCs with the BBB.

The results concerning the crossing of the BBB by MSCs in cerebral ischemia and stroke are controversial. In all studies included in Table 1, some MSCs were visualized in contact with the inner wall of the cerebral blood vessels in the first hours after IA infusion. It remains unclear whether the transplanted MSCs were actively homed [134] or they just get entrapped in the vessels [121]. Moreover, cerebral perfusion may play a significant role in stem cell distribution [135]. After the primary distribution of cells, their fate differed depending on the study. In the majority of the papers, the authors have demonstrated that MSCs can pass from the vessel lumen into the brain parenchyma within a period of time from several hours to 3 days after transplantation [123,136,137,138,139,140,141], which is in consistent with the in vitro results. However, in some studies the transplanted MSCs remained inside the cerebral blood vessels or were incorporated in the NVU at the place of pericytes, and the full crossing of the BBB by injected cells was not confirmed [124,142,143,144]. The possible reasons for this inconsistency of the results are listed below.

All studies, where BBB crossing by MSCs was not observed, had two common features: (1) transplantation of human MSCs into rats, and (2) lack of immunosuppression. If immunosuppression was performed, xenogenous human MSCs were visualized in rat brain parenchyma in the peri-infarct zone [138,139]. Notably, in both studies Cyclosporine-A has been chosen as immunosuppressive agent and this drug itself is reported to have impact on the BBB permeability (discussed in [145]). Interestingly, Kim et al. [140] observed the BBB crossing by human MSCs overexpressing neurogenin-1 after transplantation into rats without any immunosuppressive treatment, suggesting that genetic modifications may have direct or indirect impact on the ability of MSCs to pass through the BBB.

Traditionally, MSCs are considered non-immunogenic [146] and were reported to be effective in autoimmune disease treatment and even graft versus host disease due to their immunomodulatory activity [147,148,149]. However, it appears that MSCs are not completely immunoprivileged [8,150]. It was shown that MSCs transplantation in some cases can induce formation of antibodies and memory T cell; however, at a slower pace than after administration of other cell types [151,152]. Probably, the immune response to MSCs can explain why in a xenogenic host the IA transplanted MSCs persist in the brain for a short period of time and do not penetrate deep into the brain parenchyma. This hypothesis was confirmed in the work of Andrzejewska et al. [142], where human MSCs transplanted into rats with focal stroke-like focal brain injury were phagocytosed by the recipient’s activated microglia and macrophages within 3 days. The authors also reported that at least some MSCs penetrated into the perivascular space, probably contributing to the repair of the NVU by replacement of injured pericytes. Interestingly, the ability of MSCs to transmigrate across the cerebral vascular wall was also demonstrated after IA transplantation into healthy animals [153].

The long-term survival of transplanted MSCs, their crossing of the BBB and engraftment in the cerebral parenchyma may be not necessary for providing recovery after stroke. Keimpema et al. have demonstrated that stroke volume reduction can be achieved by early transient presence of intravascularly implanted MSCs at the lesion site, but not by passing of sporadic cells into the brain parenchyma [141]. In all studies presented in Table 1 where MSCs were capable of crossing the BBB, their number was small and the cells did not undergo neuronal transdifferentiation. In our recent work [144], we also did not find the transmigration of human placenta MSCs through the BBB after IA transplantation into rats with experimental stroke caused by the transient middle cerebral artery occlusion. Injected cells were transiently located inside the cerebral blood vessels in close contact with their walls for no longer than 3 days after administration, while the progressing functional recovery was observed during at least 2 weeks post-stroke. It can be speculated that MSCs may exert their positive therapeutic effects through a brief “hit and run” [151] mechanism, which triggers further prolonged molecular and cellular events. 

It is well established that infusion of MSC suspension into the brain arterial system in stroke alleviates the symptoms of the disease (discussed in [144]). The factors affecting MSCs’ distribution throughout the cerebral vascular bed after IA delivery are still not fully disclosed. Probably, the most prominent contribution is made by cerebral perfusion. It was demonstrated that distribution pattern of magnetically labeled stem cells is very similar to intra-arterial cerebral perfusion maps estimated by MRI [135]. In addition, the expression of different adhesion molecules, especially vascular cell adhesion molecule-1, on the vascular wall and chemotaxis of transplanted cells towards attractants is likely to play an important role, too [134].

The currently available evidence presented in this review suggests that the fates of the transplanted cells after they enter cerebral vascularity can be different (Figure 1). The majority of cells pass through brain capillaries and presumably go to the venous system. Some cells become transiently adherent to the small vessel walls for up to 3 days (A-MSCs in Figure 1). Still other become transiently adherent, but do not return to circulation. They pass between endothelial cells using transitory gaps formed as a result of disintegration of the tight junctions between brain endothelial cells under ischemic conditions. Then they cross the basal membrane and either enter the perivascular space (B-MSCs in Figure 1) or penetrate deeper into the brain parenchyma (C-MSCs in Figure 1). The fate of the cells crossing the BBB needs further investigation. Prompt destruction by the microglial macrophages is one of the outcomes [142]. However, most probably the extent of the phagocytosis of transplanted MSCs reaching brain parenchyma depends upon many factors and some of the cells survive. The fundamental ability of MSCs to survive in brain parenchyma is most obvious after their stereotaxic transplantation. For example, stereotaxically transplanted human placenta MSCs survive in the brain of healthy rats for at least 2 weeks [155].

The curative effects induced by cells staying inside blood vessels demonstrate that, in some cases, the paracrine mechanism may be fully responsible for the restorative action of cell therapy in stroke. In this respect, IA infusion of human MSCs into rats with experimental stroke may be a suitable in vivo model for studying the mechanisms underlying the paracrine effects of MSCs at the molecular level.

## 5. Clinical Trials of the Intra-Arterial MSC Infusion in Patients with Ischemic Stroke

Clinical trials of stem cell-based therapies for stroke have been going on for quite some time [156]. In most cases, mesenchymal stem cells, bone marrow mononuclear cells or hematopoietic stem cells were used as the cellular component of the therapy, while neural stem cells or genetically modified stem cells were used less often. Intravenous administration was the most common method of stem cell transplantation [156,157], but the ClinicalTrials.gov site reports several clinical trials where the IA stem cell infusion was utilized. In most of these trials, autologous bone marrow cells were used, and only in one trial, initiated in 2020 and still in the patient recruitment stage, allogeneic MSCs isolated from the umbilical cord are scheduled for transplantation (Table 2).

The results of Phase I and II clinical trials for the IA autologous bone marrow mononuclear cell infusion in stroke have already been published. The published trial outcome show the safety of autologous bone marrow mononuclear cells administration in sub-acute ischemic stroke [159,163], non-acute ischemic stroke [160,164], and moderate to severe acute middle cerebral artery strokes [159,165,166]. However, the analysis of the efficacy of stroke therapy with the IA injection of bone marrow mononuclear cells did not show prominent improvement in comparison with the placebo [159,163], although several trials have shown some improvement in patients receiving cell therapy [164,165]. One study showed the safety and some efficacy of intra-arterial transplantation of allogeneic umbilical cord MSCs (UC-MSC) in patients with ischemic and hemorrhagic strokes [166]. Thus, so far, clinical trials failed to fully reveal the therapeutic potential of stem cells, as it has been done in animal models of stroke. Probably, it is necessary to modify the patient recruitment criteria and optimize the treatment protocol, trying different cells, cell doses, and the frequency of transplantation, the mode and time of cell administration, and changing other parameters affecting the therapy effectiveness [157].

## 6. Conclusions

The NVU and the BBB as its part play an important role in the pathogenesis of stroke and during the post-stroke recovery.Understanding the details of interactions of transplanted MSCs with all components of the BBB and the NVU is essential for deciphering the mechanisms of MSC therapeutic effects.The results of the reviewed in vitro experiments show the ability of MSCs to pass through the monolayer of brain endothelium and substantiate the hypothesis that the BBB can be permeable for MSCs in vivo.In vitro MSCs likely pass through the monolayer of endothelium by the paracellular route using transient gaps formed in response to signals induced by ischemic conditions.Data of the in vivo experiments on transmigration of the IA transplanted MSCs across the BBB in experimental stroke are not fully consistent. The fates of transplanted cells vary: (1) Most cells just pass-through brain capillaries and presumably to the general circulation. (2) Some cells adhere to the endotheliocytes for up to several days. (3) Other cells adhere to the endothelium, then get through the endothelial layer presumably through intercellular gaps, and home in the perivascular space for no more than several days. Probably they are later destroyed by the microglial cells. (4) Cells adhere to the endothelium, pass through the BBB, and can be found in brain parenchyma for several weeks. We did not find any confirmations of their neural differentiation.Transmigration across the BBB is not necessary for induction of therapeutic effects, though, of course, it may change the parameters of the therapeutic response.Most likely, the transitory presence of the transplanted MSCs in brain blood vessels may trigger a range of therapeutic and restorative responses through paracrine secretion of an array of biologically active molecules in free form or enclosed in extracellular vesicles.Immunosuppression-free transplantation of human MSCs into rat arterial systems may be an in vivo model suitable for studying paracrine effects of MSCs in stroke.

## Figures and Tables

**Figure 1 cells-10-02997-f001:**
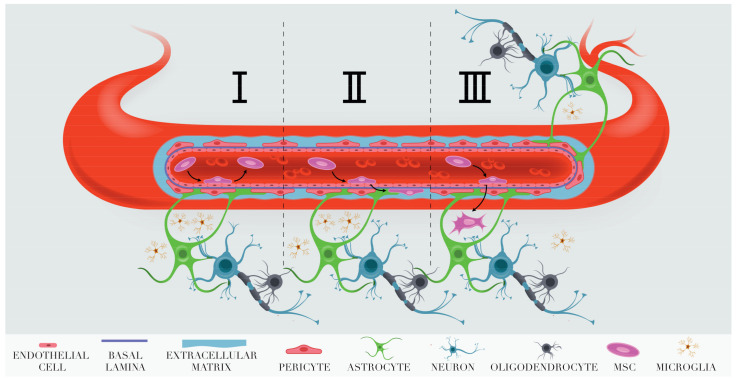
Schematic representation of the possible outcomes for MSCs adhering to the brain blood vessel endothelium after IA transplantation into rats with experimental stroke. I. Return to circulation. II. Infiltration into the perivascular space. III. Invasion into brain parenchyma.

**Table 1 cells-10-02997-t001:** Results of the studies of the penetration of MSCs through the BBB in experimental stroke models presented in selected papers. The main criterion for paper selection was quality imaging and clear description of the exact location and fate of MSCs after IA transplantation. Secondary criteria are described in the text. MCAO: middle cerebral artery occlusion. IA: intra-arterial. ICA: internal carotid artery. ECA: external carotid artery.

Study	Stroke Model	Transplanted Cell Type and Species	Cell Delivery Route and Dose	Thromboembolism Control	Immunosuppression	Time of Delivery	BBB Crossing	Location of the Transplanted Cells	Therapeutics Effects on Stroke
Walczak et al. [123]	Transient 2 h MCAO	Rat bone marrow MSCs into rats	1 × 10^6^ in 1 mL PBS, rate 1 mL/min, through the ipsilateral ICA	No	No	30 min after MCAO	Yes	1 day after transplantation MSCs were located within the brain capillaries and 10 days after in the brain parenchyma	High intracerebral engraftment correlated with significant morbidity (cerebral embolism?)
Cui et al. [136]	Transient 2 h MCAO	Integrin α4 positive rat bone marrow MSCs into rats	0.5 × 10^6^ in 0.5 mL PBS during 3 min through the ipsilateral ECA	Yes	No	24 h after MCAO	Yes	2–72 h after transplantation cells were found within the capillaries, after 72 h rare cells were located in the brain perivascular niche or parenchyma	Not described
Yavagal et al. [137]	Transient 90 min MCAO	Rat bone marrow MSCs into rats	1 × 10^6^, 5 × 10^5^, 2 × 10^5^, 1 × 10^5^, 5 × 10^4^ in 0.5 mL PBS manual injection during 3 min through the ipsilateral ICA	Yes	No	60 min and 24 h after MCAO	Yes	3–5 days after transplantation cells were located partly inside the vessels and in the adjacent brain parenchyma	Cell dose 1 × 10^5^ and below did not cause embolism; 1 × 10^5^ MSC injected 24 h after MCAO improved neurodeficit score and reduced the mean infarct volume at one month
Fukuda et al. [138]	Transient 75 min MCAO	Human bone marrow MSCs into rats	1 × 10^4^, 1 × 10^6^ in 300 μL PBS, rate 100 μL/min, through the ipsilateral ICA with maintenance of the blood flow	Yes	Cyclosporine A	24 h after MCAO	Yes	MSCs were found in the vessels’ lumen and the brain parenchyma in the peri-infarct area at 24 h post-transplantation	High- or low- dose MSCs induced behavioral recovery and microglial activation suppression at 8 days after MCAO; mortality was significantly higher in the high-dose group
Toyoshima et al. [139]	Transient 75 min MCAO	Human MSCs into rats	1 × 10^6^ in 300 μL, rate 100 μL/min, thought the ipsilateral ICA with maintenance of the blood flow	Yes	Cyclosporine-A	1, 4, 7 days after MCAO	Yes	3 h after transplantation MSCs were distributed throughout the peri-infarction zone and the infarct core, 7 days after only very few MSCs had reached the brain parenchyma	MSCs transplanted 1 and 7 days after MCAO enhanced functional recovery at 7-, 14-, and 21-days post stroke; transplantation 1 day after MCAO caused reduction in brain atrophy
Andrzejewska et al. [142]	Stroke-like focal brain injury model	ITGA4 human bone marrow MSCs into rats	5 × 10^5^ in 1 mL of PBS, rate 0.2 mL/min through the ipsilateral ICA	Yes	No	48 h after stroke modeling	No	Cells remained inside the vascular lumen over the first 2 days after IA infusion; 3 days after MSCs homed to perivascular space in the injury region; 72 h after transplantation many cells were phagocytosed	Not described
Kim et al. [140]	Transient 2 h MCAO	Human bone marrow MSCs with neurogenin 1 overexpression into rats	1 × 10^6^ in 1.2 mL saline during 5 min through the ipsilateral ICA with maintenance of the blood	No	No	3 days after MCAO	Yes	4 h after injection MSCs were mostly detected in the vascular lumen and 1 day after extravasated into the brain parenchyma	Reduction of neuronal cell death and inflammation, enhanced functional recovery in 28 days period
Toyoshima et al. [154]	Transient 90 min MCAO	Rat bone marrow MSCs into rats	1 × 10^6^ in 1 mL PBS, rate 1 mL/2 min, manual injection through the ipsilateral ICA	No	No	1 h, 6 h, 24 h, 48 h after MCAO	Yes	7 days after MCAO MSCs were mainly detected in the brain parenchyma in ischemic penumbra	24 h group displayed the best therapeutic effects: functional recovery, reduction of infarct volumes, the highest number of integrated MSCs over 7 days period
Mitkari et. [124]	Transient 90 min MCAO	Human bone marrow MSCs and pronase-detached MSCs into rats	1.1 × 10^6^, 0.5 × 10^6^ cells in 500 μL slowly through the ipsilateral ICA with maintenance of the blood	Yes	No	Acute phase after MCAO	No	Cells were entrapped within the brain capillaries immediately after transplantation and after 24 h the majority of MSCs disappeared	Not described
Keimpema et al. [141]	Transient 60 min MCAO	Rat bone marrow MSCs into rats	1 × 10^6^ in 100 μL slowly through the ipsilateral ECA	No	No	1 h after MCAO	Yes	During first 12 h MSCs were detected in the cerebral blood vessels inside and around the ischemic lesion zone. Their number started to decrease after 24 h and within 2 weeks only sporadic cells were detected in the brain parenchyma and blood vessels	A significant reduction of 50% of the ischemic lesion 2 weeks after MCAO; microglia activation
Khabbal et al. [143]	Transient 60 min MCAO	Rat and human bone marrow MSCs into rats	2 × 10^6^ in 500 µL saline during 2 min through the ipsilateral ECA with maintenance of the blood	Yes	No	24 h after MCAO	No	Both types of MSCs were located within the brain capillaries in the ipsilateral hemisphere 20 min after infusion	No data
Namestnikova et al. [144]	Transient 90 min MCAO	Human placenta MSCs into rats	5 × 10^5^ in 2 mL saline during 20 min through the ipsilateral ECA with maintenance of the blood	Yes	No	24 h after MCAO	No	MSCs were located inside the cerebral blood vessels closely sticking to their walls for no longer than 3 days after administration	Improving of the neurological deficit and survival rate of animals 14 days after transplantation

**Table 2 cells-10-02997-t002:** Clinical trials of IA MSC infusion in stroke.

No.	ClinicalTrials.gov Identifier	Title	Recruitment Status	Intervention/Treatment	Phase
1	NCT04434768	Evaluate the Safety and Explore Efficacy of Umbilical Cord Mesenchymal Stem Cells in Acute Ischemic Stroke	Recruiting	One dose of IV administration of UC-MSCs	Evaluate the Safety and Explore Efficacy of Umbilical Cord Mesenchymal Stem Cells in Acute Ischemic Stroke
2	NCT02178657 [158]	Intra-Arterial Bone-Marrow Mononuclear Cells Infusion for Acute Ischemic Stroke	Recruiting	Intra-arterial autologous bone marrow mononuclear cells injection	II
3	NCT01273337 [159]	Study of ALD-401 via Intracarotid Infusion in Ischemic Stroke Subjects	Unknown	3 mL ALDH^br^ cells (a cellular population that expresses high levels of aldehyde dehydrogenase) isolated from autologous bone marrow given as a one-time infusion via intracarotid infusion.	II
4	NCT03080571 [160]	Intraarterial Stem Cells in Subacute Ischemic Stroke	Completed	Autologous BMMNC injected in the ipsilateral MCA	I
5	NCT00473057 [160,161,162]	Study of Autologous Stem Cell Transplantation for Patients with Ischemic Stroke	Completed	Intra-arterial or intravenous delivery of autologous bone marrow cells	I

## Data Availability

Not applicable.

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
