# Peer review of "Cell Therapy of Stroke: Do the Intra-Arterially Transplanted Mesenchymal Stem Cells Cross the Blood–Brain Barrier?"

_cells, 2021, doi:10.3390/cells10112997_

Round 1

Reviewer 1 Report

The review by Yarygin et al., is a nice work describing the movement of MSCs across the blood brain barrier. The review is detailed, well presented and covers almost all aspects of the MSCs movement. The authors have excellently summarized the studies which have performed with MSCs in vivo till date. Few points the authors may consider to make the review more robust,

  1. FDA approved interventions for stroke may be included in brief. With their advantages and limitations.
  2. Authors should discuss about homing of MSCs towards the ischemic zone and the mechanism involved in their homing, the contributing factors, etc.
  3. In the title 'Blood brain barrier in stroke and its reaction to MSC transplantation', authors may consider writing 'response to MSC transplantation' instead of 'reaction to MSC transplantation'.
  4. The authors should refer to the recently published paper by Do et al., (doi: 10.3390/ijms221810045) and incorporate relevant details to make their review more updated and robust.
  5. In the figure, the arrows should be made more prominent so that one can easily understand what the authors wish to convey from their figure.

Author Response

Responses to Reviewer #1:

Dear Reviewer,

Thank you very much for reviewing our manuscript. We highly appreciate your comments and made the following amendments according to your suggestions.

  1. “FDA approved interventions for stroke may be included in brief. With their advantages and limitations”.

-We agree with this suggestion and added this information to the Introduction section. Please, see lines 39-48.

  1. “Authors should discuss about homing of MSCs towards the ischemic zone and the mechanism involved in their homing, the contributing factors, etc”.

-In the Introduction section, we introduced a paragraph in which we described the ability of MSCs to home into the damaged zone and briefly noted the mechanisms underlying this process (Please, see lines 68-85).

  1. “In the title 'Blood brain barrier in stroke and its reaction to MSC transplantation', authors may consider writing 'response to MSC transplantation' instead of 'reaction to MSC transplantation'”.

-Thank you for your comment, the title was corrected. Please, see line 149

  1. “The authors should refer to the recently published paper by Do et al., (doi: 10.3390/ijms221810045) and incorporate relevant details to make their review more updated and robust”.

-Thank you for this remark. We agree, that this new comprehensive review should be cited. We cited this paper and added additional information about the mechanisms of the BBB disruption after stroke and the effects of MSCs therapy on the restoration of the BBB in subsection 2 «Blood brain barrier in stroke and its response to MSC transplantation». Please, see lines 168-176, 193-196.

  1. “In the figure, the arrows should be made more prominent so that one can easily understand what the authors wish to convey from their figure”.

-We agree and made the arrows look more prominent.  Please, see figure 1.

Best regards, 

Irina Kholodenko, PhD

Konstantin Yarygin, MD, PhD

Reviewer 2 Report

Yarygin et al.’s review centers around the question if intra-arterially transplanted mesenchymal stem cells cross the blood-brain barrier.  The review targets an interesting question and is well written and detailed.  I have a few critical comments and some suggestions for the authors, while I appreciate the amount of thought and work that they invested in the work.

First: a very similar Review was published 2 months ago in a sister MDPI journal [1] focusing on the molecular mechanisms how the blood-brain barrier might be preserved by MSC treatment in stroke.  The authors submitted their paper before this review was published, so they should look at it and fit it into their paper, discussing the overlaps and the uniqueness of the two studies (this does not need to be long, but is necessary to be fair and up to date).

  1. In their Abstract the authors claim that “one of the most important aspects of the transplanted MSCs’ action - the ability of cells to cross the blood-brain barrier (BBB) in vitro and in vivo after IA administration into animals with experimental stroke”. This statement is not correct, since we still do not know how the successful treatment works.  It is likely one of the important factors and surely needs to be studied, but it is also possible (as the authors actually point it out in their discussion) that the cells themselves may not need to get into the parenchyma in significant number for the effect.  It might be the secretome from the cells (microvesicles and exosomes); might be how they effect the migration of immune cells from the circulation etc. So, this statement needs rewording.
  2. Introduction, line 64-65. The statement needs a reference.
  3. Throughout the paper, the authors should make it clear which species the statements (referenced work) are about. Or say for example that “unless the species is mentioned the work was done in rats”.
  4. I suggest that the authors look at the clinicaltrials.gov site and make a small Table describing the completed and ongoing trials using MSC therapy and intra-arterial infusion in stroke. There is not much there, but I feel that it is important to show that human testing is ongoing.
  5. When talking about transmigration of MSCs (starting at line 169), they do not mention the vascular muscle cells – which are also a cell layer that the MSCs would need to pass through. Furthermore, the lymphatic perivascular connection should be mentioned – there are novel human data in addition to the “glymphatic system” that also should have been mentioned. Also, they should include the transcytosis (circulating cells passing through (as opposed to in between) the vascular endothelial cells), which is known to increase in traumatic brain injury. Could it be true in stroke, too? A recent good review on the BBB by Profaci et al. [2] should be cited.
  6. It has been reported that in different CNS diseases different subpopulations of inflammatory cells can invade the CNS (for References see the Profaci review above).

Is it possible that there are certain adhesion molecules that play a role in letting circulating MSCs in the CNS in stroke or other disease? 

  1. As mentioned above, in stroke it is rather the macrophages/neutrophils that leak into the parenchyma (not lymphocytes like in MS). It has been shown that macrophages take up apoptotic MSCs which results in changing their inflammatory character to anti-inflammatory, thus alleviating inflammatory changes in the environment [3]. Could apoptotic MSCs cross into the CNS in stroke and be uptaken by activated microglia which then might revert towards the inactivated phenotype?
  2. Neutrophils make MMP9 (mentioned in the review) which is known to increase the mobility of circulating cells into the CNS [4]. MSCs can block the MMP9 synthesis in Neutrophils (see [1]) – thus MSCs might help to restore the BBB through this mechanism, as well.
  3. A minor comment: line 342 “The descriptions After the primary distribution” seems that something is missing..
  4. Finally, the Table in the MS is very hard to read. I suggest an alternative (I am giving an example below for a different format) that might make it easier on the reader to compare the different studies mentioned.

References

1          Do PT, Wu C-C, Chiang Y-H et al. Mesenchymal Stem/Stromal Cell Therapy in Blood–Brain Barrier Preservation Following Ischemia: Molecular Mechanisms and Prospects. International Journal of Molecular Sciences 2021;22(18):10045.

2          Profaci CP, Munji RN, Pulido RS et al. The blood-brain barrier in health and disease: Important unanswered questions. J Exp Med 2020;217(4).

3          Cheung TS, Galleu A, Von Bonin M et al. Apoptotic mesenchymal stromal cells induce prostaglandin E2 in monocytes: implications for the monitoring of mesenchymal stromal cell activity. Haematologica2019;104(10):e438-e441.

4          Almalki SG, Agrawal DK. Effects of matrix metalloproteinases on the fate of mesenchymal stem cells. Stem Cell Research & Therapy 2016;7(1).

Author Response

Responses to the Reviewer #2:

Dear Reviewer,

Thank you very much for thoroughly reviewing our manuscript. We highly appreciate your comments and made the following amendments according to your suggestions.

“First: a very similar Review was published 2 months ago in a sister MDPI journal [1] focusing on the molecular mechanisms how the blood-brain barrier might be preserved by MSC treatment in stroke.  The authors submitted their paper before this review was published, so they should look at it and fit it into their paper, discussing the overlaps and the uniqueness of the two studies (this does not need to be long, but is necessary to be fair and up to date)”.

-Thank you very much for this comment. This new review by Do et al. should indeed be sited and discussed in our paper. We amended section 2 «Blood brain barrier in stroke and its response to MSC transplantation» accordingly and included the reference to this review. Please, see lines 168-176, 193-196. However, we’d like to point out that in their review Do et al. focused on the MSC-based therapy aimed at the prevention of the ischemia-induced BBB compromise, while in our manuscript we analyze the ability of MSCs to cross the BBB with regard to their therapeutic effects in stroke. Therefore, our paper is intended to review different aspects of MSC-based cell therapy of stroke.

  1. “In their Abstract the authors claim that “one of the most important aspects of the transplanted MSCs’ action - the ability of cells to cross the blood-brain barrier (BBB) in vitro and in vivo after IA administration into animals with experimental stroke”. This statement is not correct, since we still do not know how the successful treatment works. It is likely one of the important factors and surely needs to be studied, but it is also possible (as the authors actually point it out in their discussion) that the cells themselves may not need to get into the parenchyma in significant number for the effect. It might be the secretome from the cells (microvesicles and exosomes); might be how they effect the migration of immune cells from the circulation etc. So, this statement needs rewording”.

-Thank you for this comment. We consider this remark to be very logical and have made the necessary changes in the text of the Abstract. Please, see lines 24-27.

  1. “Introduction, line 64-65. The statement needs a reference”.

-The references were added and the sentences were partly reformulated. Please, see lines 94-98.

  1. “Throughout the paper, the authors should make it clear which species the statements (referenced work) are about. Or say for example that “unless the species is mentioned the work was done in rats””.

-We agree with this important remark and added missing information in the subsections 2 and 4.

  1. “I suggest that the authors look at the clinicaltrials.gov site and make a small Table describing the completed and ongoing trials using MSC therapy and intra-arterial infusion in stroke. There is not much there, but I feel that it is important to show that human testing is ongoing”.

-We have made a small section in the manuscript devoted to clinical trials of intra-arterial administration of MSCs in stroke. We also posted a table with data from clinicaltrials.gov there. Please see section 5. Clinical trials of the intra-arterial MSC infusion in patients with ischemic stroke.

  1. “When talking about transmigration of MSCs (starting at line 169), they do not mention the vascular muscle cells – which are also a cell layer that the MSCs would need to pass through. Furthermore, the lymphatic perivascular connection should be mentioned – there are novel human data in addition to the “glymphatic system” that also should have been mentioned. Also, they should include the transcytosis (circulating cells passing through (as opposed to in between) the vascular endothelial cells), which is known to increase in traumatic brain injury. Could it be true in stroke, too? A recent good review on the BBB by Profaci et al. [2] should be cited”.

-We mentioned in the text the vascular muscle cells (Please, see lines 239) Briefly described their role in maintaining the integrity of the BBB in health and in stroke (Please, see lines 257-266).

-We also mentioned the role of the “glymphatic” system in maintaining the integrity of the BBB and the effect of MSC transplantation on the “glymphatic” system (Please, see lines 214-225).

- The transcytosis as the possible mechanism of MSCs’s migration across the BBB has been already mentioned in the subsection 3. Thank you for the recommendation of the recent review by Profaci et al. We cited this paper in the subsection 2. Please, see lines 163.

  1. “It has been reported that in different CNS diseases different subpopulations of inflammatory cells can invade the CNS (for References see the Profaci review above)”.

-Thank you for this recommendation. In the detailed retain and well-illustrated review by Profaci et al. there is comprehensive information about the stricture, function and mechanism of damage of the BBB. We added the information to section 2 and cited Profaci’s paper. Please, see lines 163

“Is it possible that there are certain adhesion molecules that play a role in letting circulating MSCs in the CNS in stroke or other disease?”

-Yes, this mechanism has been demonstrated in several studies in experimental stroke models. We added the information about factors, that could potentially influence stem cells distribution to section 4. Please, see lines 476-483.

  1. “As mentioned above, in stroke it is rather the macrophages/neutrophils that leak into the parenchyma (not lymphocytes like in MS). It has been shown that macrophages take up apoptotic MSCs which results in changing their inflammatory character to anti-inflammatory, thus alleviating inflammatory changes in the environment [3]. Could apoptotic MSCs cross into the CNS in stroke and be uptaken by activated microglia which then might revert towards the inactivated phenotype?”

- Thank you for providing us with this interesting reference giving new insights in the MSCs’ mechanisms of action. We discussed this theory in introduction (Please, see lines 111-118). However, this theory still must be proved in experimental stroke studies.

  1. “Neutrophils make MMP9 (mentioned in the review) which is known to increase the mobility of circulating cells into the CNS [4]. MSCs can block the MMP9 synthesis in Neutrophils (see [1]) – thus MSCs might help to restore the BBB through this mechanism, as well”.

-We have described such a mechanism for the positive effect of MSCs on the BBB restoration, using the recommended references (Please, see lines 226-236).

  1. “A minor comment: line 342 “The descriptions After the primary distribution” seems that something is missing”.

-We corrected this typo (deleted «The descriptions»). Please, see line 425.

  1. “Finally, the Table in the MS is very hard to read. I suggest an alternative (I am giving an example below for a different format) that might make it easier on the reader to compare the different studies mentioned”.

-We agree with this comment and tried table reformatting to make it easier for the readers to understand the content of the table. Please see corrected table 1.

Best regards, 

Irina Kholodenko, PhD

Konstantin Yarygin, MD, PhD

Round 2

Reviewer 2 Report

Thank you for following my suggestions and I accept all the changes you made.  I have one remaining suggestion.  You added the paragraph about the "glymphatic" system; most of those works are based on rodent data. I feel that you should cite the recent paper that specifically talks about the lymphatic elements in human brains (PMID: 33446503), and add a sentence about the findings.

Author Response

Dear Reviewer,

Thank you very much for your comment. We got acquainted with the work recommended by you and entered the corresponding text into the manuscript (Please, see lines 225-239).

Best regards, 

Irina Kholodenko, PhD

Konstantin Yarygin, MD, PhD